# Glucose Deprivation Induces Cancer Cell Death through Failure of ROS Regulation

**DOI:** 10.3390/ijms241511969

**Published:** 2023-07-26

**Authors:** Mingyu Kang, Joon H. Kang, In A. Sim, Do Y. Seong, Suji Han, Hyonchol Jang, Ho Lee, Sang W. Kang, Soo-Youl Kim

**Affiliations:** 1Division of Cancer Biology, Research Institute, National Cancer Center, Goyang 10408, Gyeonggi-do, Republic of Korea; 75957@ncc.re.kr (M.K.); wnsl2820@ncc.re.kr (J.H.K.); siminae@ncc.re.kr (I.A.S.); ds528491@gmail.com (D.Y.S.); ho25lee@ncc.re.kr (H.L.); 2New Cancer Cure Bio Co., Goyang 10408, Gyeonggi-do, Republic of Korea; 3Division of Rare and Refractory Cancer, Research Institute, National Cancer Center, Goyang 10408, Gyeonggi-do, Republic of Korea; hanbonoboss@ncc.re.kr (S.H.); hjang@ncc.re.kr (H.J.); 4Graduate School of Cancer Science and Policy, National Cancer Center, Goyang 10408, Gyeonggi-do, Republic of Korea; 5Department of Life Science, Ewha Women’s University, Seoul 03760, Republic of Korea; kangsw@ewha.ac.kr

**Keywords:** cancer metabolism, glycolysis, glucose deprivation, ROS, cell death

## Abstract

In previous work, we showed that cancer cells do not depend on glycolysis for ATP production, but they do on fatty acid oxidation. However, we found some cancer cells induced cell death after glucose deprivation along with a decrease of ATP production. We investigated the different response of glucose deprivation with two types of cancer cells including glucose insensitive cancer cells (GIC) which do not change ATP levels, and glucose sensitive cancer cells (GSC) which decrease ATP production in 24 h. Glucose deprivation-induced cell death in GSC by more than twofold after 12 h and by up to tenfold after 24 h accompanied by decreased ATP production to compare to the control (cultured in glucose). Glucose deprivation decreased the levels of metabolic intermediates of the pentose phosphate pathway (PPP) and the reduced form of nicotinamide adenine dinucleotide phosphate (NADPH) in both GSC and GIC. However, glucose deprivation increased reactive oxygen species (ROS) only in GSC, suggesting that GIC have a higher tolerance for decreased NADPH than GSC. The twofold higher ratio of reduced/oxidized glutathione (GSH/GSSG) in GIS than in GSC correlates closely with the twofold lower ROS levels under glucose starvation conditions. Treatment with N-acetylcysteine (NAC) as a precursor to the biologic antioxidant glutathione restored ATP production by 70% and reversed cell death caused by glucose deprivation in GSC. The present findings suggest that glucose deprivation-induced cancer cell death is not caused by decreased ATP levels, but rather triggered by a failure of ROS regulation by the antioxidant system. Conclusion is clear that glucose deprivation-induced cell death is independent from ATP depletion-induced cell death.

## 1. Introduction

In previous work, we showed that glucose deprivation for 24 h does not change ATP levels in various cancer cell lines (glucose insensitive cancer cells, GIC) [1,2]. Cancer cells utilize fatty acids from the blood for ATP production through a catabolic process that involves fatty acid oxidation (FAO), the TCA cycle, the electron transfer chain (ETC), and oxidative phosphorylation (OxPhos) [1]. However, in a population of glucose sensitive cancer cells (GSC), ATP production decreases significantly within 12 h of glucose deprivation. Glucose deprivation-induced cell death is traditionally explained as a cascade of reactions involving the inactivation of glycolysis and decreased ATP production leading to cell death; this is based on the notion that glycolysis is the major route for ATP production in cancer cells [3,4]. However, the mechanism of glucose deprivation-induced cell death has not been investigated in detail in relation to cancer metabolism. Several mechanisms of glucose deprivation-induced cell death have been proposed, including increased ER stress [5], caspase 8 activation [6], and AMPK-induced autophagic cell death [7]. Although these theories do not provide evidence that decreased ATP production precedes cell death, a reduction in ATP levels is considered as a key inducer of cell death in cancer cells deprived of glucose because, according to the Warburg effect, ATP production absolutely relies on glycolysis in cancer cells.

Because ATP levels in GSC decrease within 12 h of glucose deprivation concomitant with an increase in cell death, in this study we investigated whether the decrease in ATP production upon glucose deprivation is a cause or a result of cell death in GSC. 

## 2. Results

### 2.1. ATP Production Decreases in Some Cancer Cells after 24 h of Glucose Deprivation 

Previously, we reported that in various cancer cell lines, including pancreatic cancer, ovarian cancer, lung cancer, colon cancer, and GBM cells, ATP production does not decrease in response to glucose deprivation for 24 h, whereas lactate production stops [1]. Glucose deprivation did not change ATP levels in pancreatic cancer cell lines such as AsPC-1 cell and Panc-1 cells at 12 h, consistent with previous results [1] (Figure 1A,B). These cells were grouped as GIC. However, in some cancer cell lines, including glioma cells (U87 and T98G), breast cancer cells (MDA-MB-231 and MCF-7), and colon cancer cells (KM12 and HT-29), glucose deprivation for 12 h decreased ATP production by 20–80% compared with the control cells cultured with high glucose (Figure 1C,D and Appendix A). These cells were grouped as GSC. This effect was traditionally considered to be the result of a decrease in glycolysis metabolism, which is a major ATP production pathway. Furthermore, studies show that glucose depletion induces cell death [5,8]. Therefore, it was accepted universally that blocking glycolysis induces cancer cell death by ATP depletion. However, this explanation of the phenomenon is inconsistent. First, ATP depletion does not induce apoptosis immediately and instead induces cell cycle arrest [9]. Second, there are no reports showing a queue of cell death including ATP depletion by glucose deprivation. Cell death can be induced by glucose starvation without proof of ATP depletion [5,6,7]. Third, this explanation contradicts our theory of FAO dependency, which was demonstrated in various cancer cells under glucose deprivation [1]. Therefore, whether the decrease in ATP production is a cause or a result of cell death remains an unanswered question. 

### 2.2. Glucose Deprivation Induces Cell Death in GSC within 12 h

To determine whether decreased ATP production is related to cell death, cell death was measured at 12 and 24 h. Glucose deprivation did not increase cell death in AsPC-1 cells at 24 h (Figure 2A,B), which is consistent with previous results in GIC [1]. However, in GSC, cell death increased by twofold at 12 h and by up to eightfold at 24 h according to FACS analysis of annexin V staining (Figure 2C,D and Appendix A). It remains unclear whether the 20–80% decrease in ATP production shown in Figure 1C,D is the result of decreased glycolysis or cell death induced by glucose deprivation.

### 2.3. ATP Levels in GSC and GIC do Not Change Markedly after 6 h of Glucose Deprivation 

Metabolic changes including changes in ATP levels in GSC and GIC were analyzed by mass spectrometry at 3 and 6 h after glucose deprivation (Figure 3). In the GSC cell line PC-3, lactate production decreased by 60–80% compared with that in the control cultured in a high glucose medium, indicating a decrease in the rate of glycolysis of >60% (Figure 3A,B). Glucose deprivation caused a rapid decrease in lactate levels by approximately 40% at 1 h in both GSC (PC-3) and GIC (AsPC-1) (Appendix A). A 60–80% decrease in 6-phospho-gluconate (6-PG) compared with the control indicated a decrease in pentose phosphate pathway (PPP) metabolism of >60% in 6 h (Figure 3A,B). Glucose-6-phosphate dehydrogenase (G6PD) activity decreased by approximately 40% within 1 h in both GSC and GIC (Appendix A). However, no changes in ATP levels were observed at 6 h in GSC and GIC (Figure 3A,B and Appendix A). Metabolic analysis showed that glucose deprivation caused a consistent decrease of glycolysis intermediates in GSC and GIC.

### 2.4. Glucose Deprivation Induces Cell Death Only in GSC with a Significant Increase in ROS Production

Glucose deprivation decreased lactate production by 70% at 24 h in AsPC-1 (GIC) and PC-3 (GSC) cells (Figure 4A). The activity of G6PD, a key PPP enzyme, decreased by 40–50% in AsPC-1 (GIC) and PC-3 (GSC) cells (Figure 4B), suggesting a 40–50% decrease in PPP metabolism caused by glucose deprivation at 12 h. NADPH production also decreased to 50–60% of the control in both AsPC-1 (GIC) and PC-3 (GSC) cells at 24 h (Figure 4C). ROS levels did not change in AsPC-1 (GIC) cells (Figure 4D), whereas they increased by approximately 70% in PC-3 (GSC) cells exposed to glucose deprivation for 24 h (Figure 4E). GSC (U87, MDA-MB-231, and KM12 cells) also showed a significant increase in ROS levels at 24 h (Appendix A). Therefore, glucose deprivation-induced cell death was correlated with an increase in ROS production in PC-3 cells (Figure 2C,D), whereas in AsPC-1 cells, glucose deprivation did not induce cell death or increase ROS, but it inhibited glycolysis and decreased NADPH production from the PPP at 24 h (Figure 1A). These results indicate that ROS tolerance differs markedly between GSC and GIC. GIC showed a higher tolerance to decreased NADPH caused by glucose deprivation than GSC. ROS regulation is a complex mechanism because ROS producers and scavengers need to be balanced to achieve tolerance. An important ROS regulator is glutathione (GSH), which is the most abundant antioxidant in cancer cells and protects redox-sensitive proteins under stress conditions [10]. In the presence of high glucose levels, the cellular GSH/GSSHR ratio did not differ between GIC (AsPC-1, SNB-75, and SK-OV-3) and GSC (KM12, PC-3, and MDA-MB-231), whereas under glucose deprivation conditions, the GSH/GSSG ratio was twofold higher in GIC than in GSC (Figure 4F and Appendix A). Therefore, a higher GSH/GSSG ratio in GIC than in GSC may contribute to ROS tolerance. 

### 2.5. Glucose Deprivation-Induced GSC Cell Death Is Rescued by the Antioxidant NAC 

Experimental data from glucose deprivation experiments suggest that the decrease in ATP production was not induced by decreased glycolysis (Figure 3) but by an increase in ROS regulation (Figure 4). To determine whether the decrease in ATP was the result of increased ROS production and ROS-mediated cell death, GSC were treated with the antioxidant N-acetyl cysteine (NAC) [11] under glucose deprivation conditions. Glucose deprivation caused a 20–30% increase in ROS levels in PC-3 and DU-145 cells (GSC) (Figure 5A,B) and other GSC cells (U87, MDA-MB0231, KM12) (Appendix A). Treatment with 5 mM NAC for 12 h decreased ROS production to control levels (Figure 5A,B). In addition, NAC restored ATP production to 70% of the control in PC-3 and DU-145 cells (GSC) (Figure 5C,D) and to 70–90% of the control in GSC (U87, MDA-MB0231, KM12) (Appendix A). NAC treatment also restored glucose deprivation-induced cell death, resulting in fourfold and twofold reductions in cell death in PC-3 and DU-145 cells, respectively (Figure 5E,F). 

To investigate the temporal relationship between changes in ROS levels and cell death upon glucose deprivation at the single-cell level, live cell imaging was performed using the real-time live cell imager Operetta CLS. In AsPC-1 cells, glucose deprivation did not affect the levels of ROS or cell death up to 24 h (Figure 6A–D). However, in PC-3 cells, ROS levels began to significantly increase after 12 h of glucose deprivation, followed by an increase in cell death at 18 h compared with the control (Figure 6E–G). Approximately 80% of dead cells showed a significant increase in PI staining following the marked increase in ROS levels in PC-3 cells (Figure 6E). Treatment with NAC prevented the increase in ROS levels and significantly reduced cell death in PC-3 cells (Figure 6E,H). These findings suggest that the increase in ROS levels plays a key role in mediating cell death under glucose deprivation conditions in GSC.

## 3. Discussion

Glucose is considered as a major source of ATP in cancer cells through glycolysis according to the Warburg effect. However, glucose deprivation for 24 h does not change ATP levels despite the inhibition of lactate production in various cancer cells [2]. We found that the major source of ATP production in cancer is FAO in a process named the ‘Kim effect’ to avoid confusion with metabolic reprogramming [1]. These controversies can be explained by differences in the culture media. In 1924, Dr. Warburg measured oxygen consumption using tumor slices incubated in isotonic Ringer solution containing only glucose [12]. Therefore, the tumor could not consume oxygen for OxPhos because glucose was the only carbon source metabolized into lactate. Later, in 1959, Minimal Essential Medium (MEM), developed by Harry Eagle [13], was introduced as a synthetic cell culture medium containing amino acids and vitamins. In 1958, a culture medium supplemented with FBS was developed by Arthur Robinson [14]. Current cell culture media such as RPMI and DMEM mimic human plasma by including glutamine, pyruvate, bicarbonate, and glucose, in addition to supplementation with 10% FBS. Therefore, under glucose deprivation conditions, cancer cells can reroute the TCA pathway, ETC, and OxPhos activity using bio building blocks such as fatty acids, amino acids, and pyruvate, which results in the consumption of oxygen. Cancer cells rely on FAO for the supply of TCA intermediates as well as ATP production through OxPhos, as evidenced by blocking FAO, which reduces OxPhos and ATP production under high glucose conditions and in the presence of supplements including amino acids, pyruvate, and FBS [1,2].

Under glucose deprivation conditions, ROS-mediated cell death increased in GSC, and this was caused by the suppression of the PPP and the resulting depletion of NADPH. In cancer cells, glucose is metabolized to stimulate the PPP to produce pentose phosphate for nucleic acid synthesis and NADPH for ROS regulation, TCA cycle, and fatty acid synthesis [15]. We showed that glucose deprivation for 3 h decreased the PPP intermediate ribose-5-phosphate to 20% of the control under normal glucose conditions in pancreatic cancer cells according to mass analysis (Figure 3), which is consistent with previous observations [2]. The PPP generates pentose phosphate from glucose in parallel with glycolysis producing lactate. Therefore, glucose deprivation results in the depletion of lactate as well as PPP intermediates. ROS are balanced by reducing factors such as NADPH to promote cancer growth [16]. When ROS levels exceed the regulation threshold, ROS induce the regulated cell death (RCD) program such as apoptosis, necroptosis, and ferroptosis [17]. ROS accelerate apoptosis by promoting the ubiquitin-mediated proteasomal degradation of anti-apoptotic factors such as c-FLIP [18]. ROS and necroptosis can form a positive feedback loop. Increased ROS oxidize RIP1, causing its self-activation and protecting RIP3 from caspase 8-mediated cleavage [19]. RIP3 also promotes TCA cycle activity and aerobic respiration, which increase ROS generation [19]. Ferroptosis, a unique cell death pathway that is distinct from apoptosis and necrosis, is an iron- and ROS-dependent form of RCD [20]. In this study, the reducing agent NAC rescued GSC from glucose deprivation-induced cell death (Figure 5). This implies that decreased ATP production is not the cause of cell death, but rather a result of cell death caused by failure of ROS regulation. ROS have dual and opposite roles in cancer, as they can induce cell proliferation to promote cancer growth but can also have a tumor suppression effect by activating RCD programs such as apoptosis, necroptosis, and ferroptosis [21]. ROS levels are higher in cancer than in normal cells because cancer cells have a higher content of antioxidants [22]. The higher levels of ROS in cancer cells play critical roles in the activation of cell signaling and epithelial-mesenchymal transition [23], the activation of transcription factors such as Nrf2 [24], and the induction of lipid peroxidation products such as 4-hydroxy-2-nonenals [25], which can produce NADH for ATP production by aldehyde dehydrogenase [26]. However, increased ROS levels also induce cancer cell death [21]. Cancer cell death is caused by glucose depletion [8] or glycolytic enzyme knockdown, as well as by chemotherapy or radiation therapy which is closely associated with induced ROS. Therefore, a role of ROS in anticancer therapies was proposed [27]. In response to stress related to ROS induction, antioxidants such as NAC promote cancer survival and progression, which is consistent with the data presented in Figure 5E,F [27]. 

Glucose deprivation-induced cell death is traditionally explained as the result of decreased ATP production based on the Warburg effect, which proposes that glycolysis is the major metabolic pathway for ATP production (Figure 7A). Several mechanisms of ATP deprivation-induced cell senescence or cell death have been proposed, including cell cycle arrest [9], increased ER stress [5], caspase 8 activation [6], increased protein aggregation [28], and AMPK-induced autophagic cell death [7,29] among others. However, there are no experimental data demonstrating that glucose deprivation induces cell death by decreasing ATP production. ATP depletion also induces apoptosis. In studies of energy metabolism, oligomycin treatment targeting OxPhos by inhibiting ATP synthase decreases ATP production and induces cell death [30]. Blocking OxPhos results in cell death after AMPK activation, ROS generation, and caspase activation despite normal glycolytic activity [30]. Oligomycin treatment under glucose deprivation conditions reduces ATP levels by >50% within 10 min [31]. However, cell death is not an immediate response to ATP depletion and is induced after 48 h [30]. ATP depletion may lead to multiple metabolic alterations of cell homeostasis before cell death occurs. Indeed, glucose deprivation-induced cell death may be independent from ATP depletion-induced cell death. In this study, glucose deprivation significantly decreased NADPH production in cancer cells by blocking the PPP. The PPP is important for ribonucleotide synthesis and NADPH production in cancer and in normal cells. However, cancer cells require high levels of ribonucleotide production for growth and NADPH for redox balance [15]. To meet this demand, cancer activates the PPP through glycolysis for NADPH production. NADPH is required as a cofactor for many important physiological reactions involved in cancer growth, including fatty acid synthesis, folate metabolism [16], and reducing ROS levels [16]. Among them, an increase in ROS levels is an important effect of cell death caused by glucose deprivation, which is directly linked to a decrease of NADPH production from the PPP. Therefore, many clinical trials targeting NADPH metabolic enzymes such as isocitrate dehydrogenase are ongoing [32]. In cancer cells, the PPP plays a key role in NADPH production because the PPP is a defense system in normal cells. In normal skin cells, the PPP is activated as a first-line response to oxidative stress [33]. In addition to its effect on increasing ROS production, NADPH depletion may induce cell death by decreasing dihydrofolate reductase (DHFR) activity, thereby impairing the one-carbon biosynthesis pathway [16]. DHFR converts dihydrofolate into tetrahydrofolate, which is required for the de novo synthesis of purines and thymidylic acid. Pemetrexed is an inhibitor of DHFR that is approved for the treatment of cancer [34]. Therefore, we cannot disregard the NADPH requirement for various enzyme reactions.

Recently, we proposed that a major metabolic pathway of ATP production in cancer is FAO using fatty acids systemically supplied from the blood system (Kim effect) (Figure 7B,C) [1]. This was based on results showing that ATP levels do not decrease in various cancer cells exposed to glucose deprivation for 24 h [1,2]. Cancer cells that did not show changes in ATP in response to glucose deprivation were called GIC (Figure 7C) in contrast to GSC (Figure 7B). However, some cancer cells were sensitive to glucose deprivation, which induced cell death by increasing ROS production through the downregulation of NADPH, resulting in decreased ATP production (Figure 7B). Although glucose deprivation decreased NADPH production in both GIC and GSC through the depletion of PPP intermediates, ROS levels increased in GSC but not in GIC (Figure 7C). The exact mechanism of ROS regulation in GIC needs to be investigated further. In summary, we conclude that ATP depletion by glucose deprivation is the result of cancer cell death caused by failure of ROS regulation by the antioxidant system. Indeed, glucose deprivation-induced cell death is independent from ATP depletion-induced cell death.

## 4. Materials and Methods

### 4.1. Cell Culture

Human cancer cell lines were obtained from American Type Culture Collection (ATCC) and Korean Cell Line Bank. Cells were incubated at 37 °C and maintained in 5% CO_2_. Panc-1 (pancreatic cancer), and U87 and T98G (glioblastoma multiforme) cells were grown in high glucose DMEM (SH30243.01; Hyclone, Logan, UT, USA) containing 10% fetal bovine serum (FBS; SH30070.03HI, HyClone), penicillin, and streptomycin. KM-12 and HT-29 (colon cancer); PC-3 and DU-145 (prostate cancer); MDA-MB-231 and MCF-7 (breast cancer); and AsPC-1 (pancreatic cancer) cell lines were grown in RPMI 1640 medium (SH30027.01, HyClone) containing 10% FBS, penicillin, and streptomycin. 

### 4.2. Seahorse Mito Stress Test Assay

To test the effect of glucose on cellular respiration in cancer cells, 1–2 × 10^4^ cells were seeded in each well of a seahorse microplate, and after 24 h, cells were treated with or without glucose medium (11966025 and 111879020) for 24 h. For determination of oxygen consumption rate (OCR), cells were incubated in XF base medium supplemented with 0 and 10 mM glucose, 1 mM sodium pyruvate, and 2 mM L-glutamine. Then, cells were equilibrated in a non-CO_2_ incubator for 1 h before starting the assay. During the incubation, the mitochondrial inhibitors oligomycin (1 µM), FCCP (0.5 µM), and rotenone/antimycin A (0.5 µM) dissolved in XF base medium were injected at the XFe96 sensor cartridge. Finally, normalization was performed with the SRB assay.

### 4.3. FITC Annexin V and Propidium Iodide (PI) Cell Death Detection

Cell death was analyzed using the annexin V-AbFlour^TM^ 488 apoptosis detection kit (KTA0002, Abbkine scientific, Wuhan, China). Cells were cultured for 0, 12, and 24 h under glucose-free conditions. Cells were collected, washed with cold PBS twice, centrifuged at 4 °C for 3 min, and resuspended in 100 µL of 1× annexin V binding buffer to a concentration of 2 × 10^5^ cells/mL. The solution (100 µL) was mixed gently with 4 µL of annexin V-AbFlour^TM^ 488 and 1 µL of PI. The cells were incubated for 15 min at room temperature in the dark, and 400 µL of 1× annexin V binding buffer was added to each tube. The samples were analyzed using a FACSVerse flow cytometer (BD Falcon, Bedford, MA, USA). All apoptotic cell death rates (Annexin V-positive, PI-positive, and double-positive) were quantified using FlowJo software (v10.8.1).

### 4.4. Reactive Oxygen Species (ROS) Measurement

Cellular reactive oxygen species (ROS) were determined using the DCFDA/H2DCFDA—Cellular ROS Assay kit (ab113851, Abcam, Cambridge UK). Trypsinized cells (2–5 × 10^5^) were washed with PBS and incubated with 20 µM of the ROS indicator 2,7-dichlorofluorescin diacetate (DCFDA) in 1× buffer at 37 °C for 30 min. The levels of cellular ROS were analyzed using the BD FACSVerse^TM^ flow cytometer. Fluorescence intensity was quantified by the Geometric Mean in FlowJo software (v10.8.1)**.**

### 4.5. Lactate Measurement 

Lactate levels were measured using the L-Lactate Assay Kit (ab65330, Abcam). Trypsinized cells (1–2 × 10^6^) were washed with cold PBS, resuspended in lactate assay buffer, and centrifuged at 4 °C at top speed for 5 min. The supernatants were collected, mixed with 50 µL of reaction reagents, and incubated at room temperature for 30 min. The absorbance was measured using a microplate reader at OD 570 nm.

### 4.6. ATP Measurement 

ATP levels were measured using an ATP Assay Kit (ab83355, Abcam). Trypsinized cells were washed with cold PBS, lysed in ATP assay buffer, and centrifuged at 4 °C at top speed for 2 min. The supernatants were collected and mixed with 50 µL of reaction reagents before incubation at room temperature for 30 min. The absorbance was measured using a microplate reader at OD 570 nm.

### 4.7. NADPH Assay

The NADPH/NADP ratio was determined using a NADPH Assay Kit (ab65349, Abcam). Trypsinized cells were lysed in 800 μL of assay buffer and homogenized with two freeze/thaw cycles (20 min on dry ice followed by 10 min at room temperature). Samples were vortexed and centrifuged at top speed for 5 min. Then, samples were passed through a needle-fitted syringe to shear DNA. To detect NADPH, NADP needs to be decomposed before the reaction. Aliquots containing 200 μL of extracted samples were placed into new e-tubes and heated to 60 °C for 30 min, and 50 μL of sample was added to each well of a 96-well plate. A reaction mix containing 98 μL NADPH Cycling Buffer and 2 μL of NADPH Cycling Enzyme Mix was made, and 100 μL of the reaction mix was added to each well containing a test sample. Then, the plate was incubated at room temperature for 5 min and 10 μL NADPH Developer was added into each well. The plate was then incubated at room temperature for 1–4 h. The absorbance was measured using a microplate reader at OD 450 nm.

### 4.8. GSH/GSSG Ratio

The GSH/GSSG ratio was determined using the EZ-Glutathione Assay Kit. Trypsinized cells were washed in cold PBS twice, suspended in cold 5% MPA, and homogenized by sonication. Samples were centrifuged at 4 °C at 12,000 rpm for 10 min. To measure GSH and GSSG levels, the collected supernatants were mixed with reaction reagents in a 96-well plate separately. The plate was incubated at room temperature for 5 min and 50 μL NADPH was added to each well. After the reaction, the absorbance was measured using a microplate reader at OD 412 nm.

### 4.9. LC-MS/MS

Metabolites related to energy metabolism were analyzed with an LC-MS/MS system equipped with the 1290 HPLC system (Agilent, Waldbronn, Germany) and QTRAP 5500 (AB Sciex, Toronto, ON, Canada) and a reverse phase column (Synergi fusion RP 50 × 2 mm). Mobile phases A and B consisted of 5 mM ammonium acetate in H_2_O and 5 mM ammonium acetate in methanol, respectively. The separation gradient was as follows: hold at 0% B for 5 min, 0% to 90% B for 2 min, hold at 90% for 8 min, 90% to 0% B for 1 min, then hold at 0% B for 9 min. The LC flow was 70 μL/min and 140 μL/min between 7 and 15 min and the column temperature was kept at 23 °C. Multiple reaction monitoring (MRM) was used in negative ion mode, and the extracted ion chromatogram (EIC) corresponding to the specific transition for each metabolite was used for quantification. The area under the curve for each EIC was normalized to that of the EIC of the internal standard. The peak area ratio of each metabolite to the internal standard was normalized to the protein amount. Data analysis was performed using Analyst 1.7.1 software.

### 4.10. Real-Time Live Cell Imaging and High Content Analysis

Cells (1.5 × 10^4^) were seeded in a PhenoPlate 96-well (PerkinElmer, Waltham, MA, USA) and incubated at 37 °C. After one day, cells were stained with 1 µg/mL PI and 5 µM CellROX Green Reagent (Thermo Fisher Scientific, Waltham, MA, USA) for detection of cell death and oxidative stress. Groups of cells were incubated with N-acetyl cysteine (NAC, 5 mM) for 1 h. After 1 h, the medium was removed, and cells were washed with PBS twice. The cell culture medium was changed to glucose-free medium containing all dyes with or without NAC. Real-time live cell images were acquired using the Operetta CLS (Perkin Elmer, Waltham, MA, USA), capturing bright-field and fluorescent images every hour from 0 to 24 h, as previously described [35]. For each time point, nine field images were taken per well, and a minimum of 1 × 10^4^ cells were analyzed per well. High content analysis was performed using Harmony 4.5 software (Perkin Elmer). The cell count and fluorescent intensity for each cell were quantified in the images at each time point. Intensity values higher than those of the control cells were considered as positive signals. Cells that exhibited ROS-positive signals prior to the appearance of PI-positive signals were classified into the ‘PI positive after ROS positive’ group. Cells that exhibited only PI-positive signals without showing preceding ROS-positive signals were classified into the ‘PI positive after ROS negative’ group.

### 4.11. Statistical Analysis 

The experimental data are presented as the mean ± SD. All statistical analyses were performed using the GraphPad Prism 9 software (GraphPad Software Inc., San Diego, CA, USA). Statistical significance was analyzed with a one-way analysis of variance (ANOVA). *p*-values are denoted as follows: * *p* < 0.05, ** *p* < 0.01, *** *p* < 0.001

## 5. Conclusions

In this study, we found that glucose deprivation triggers cell death following an increase in ROS levels that was inversely correlated with ATP production. The increase of ROS induced cell death was concomitant with a decrease in ATP levels. Treatment with the reducing agent NAC reversed cancer cell death after restoration of ATP levels in cells under glucose deprivation. These findings suggest that ATP depletion resulting from glucose deprivation is not the cause of cancer cell death, but rather the result of cancer cell death caused by failure of ROS regulation. Indeed, glucose deprivation-induced cell death is independent from ATP depletion-induced cell death.

## Figures and Tables

**Figure 1 ijms-24-11969-f001:**
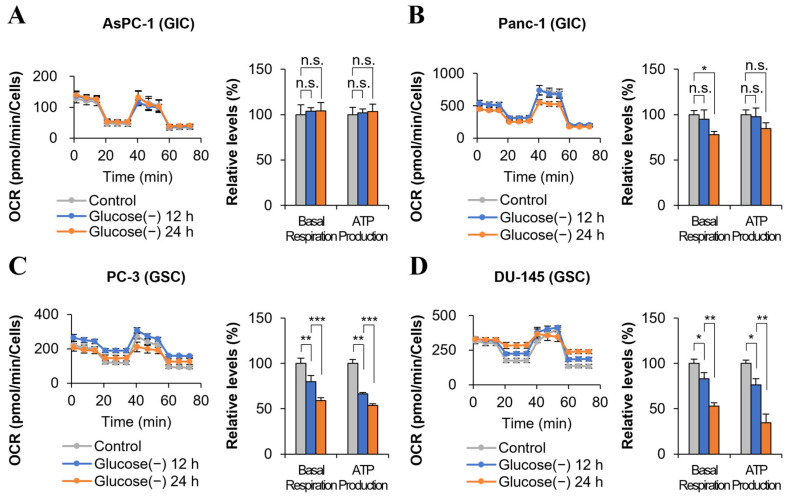
Some glucose sensitive cancer cells show reduced ATP production under glucose deprivation conditions. (**A**,**B**) Basal respiration and ATP production were measured by XFe96 extracellular flux analysis in pancreatic cancer cells [glucose insensitive cells (GIC): AsPC-1 and Panc-1] after glucose deprivation for 0, 12, and 24 h (AsPC-1, n = 5; Panc-1, n = 4). (**C**,**D**) Basal respiration and ATP production were measured by XFe96 extracellular flux analysis in prostate cancer cells [glucose sensitive cells (GSC): PC-3 and DU-145] after glucose deprivation for 0, 12, and 24 h (PC-3, n = 4; DU-145, n = 4). Glucose (-), glucose deprivation; graph bars represent the mean ± sd. *, *p* < 0.05; **, *p* < 0.01; ***, *p* < 0.001; n.s., not significant.

**Figure 2 ijms-24-11969-f002:**
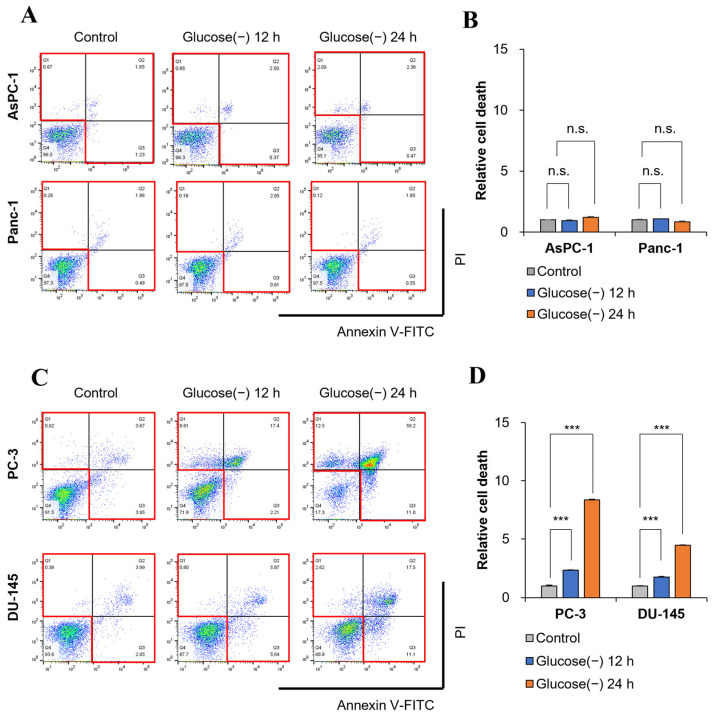
Glucose deprivation induces apoptosis in GSCs. (**A**,**B**) Cell death was measured by annexin V-FITC and propidium iodide (PI) staining in pancreatic cancer cells (GICs) of AsPC-1 and Panc-1 cultured under glucose deprivation conditions for 0, 12, and 24 h (n = 3). (**C**,**D**) Cell death (red box)was measured by annexin V-FITC and PI staining in prostate cancer cells (GSCs) of PC-3 and DU-145 cultured under glucose deprivation conditions for 0, 12, and 24 h (n = 3). Apoptotic rates were quantified using FlowJo software (v10.8.1). Glucose (-), glucose deprivation; graph bars represent the mean ± sd. ***, *p* < 0.001; n.s., not significant.

**Figure 3 ijms-24-11969-f003:**
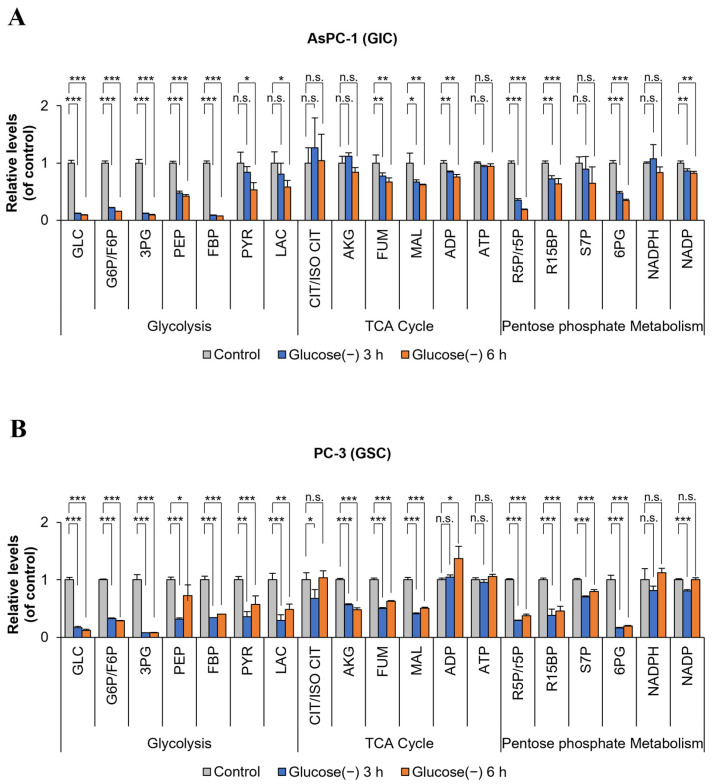
Metabolic analysis of GICs and GSCs cultured under glucose starvation conditions for 0, 3, and 6 h. (**A**,**B**) Metabolites of glycolysis, the TCA cycle, and the pentose phosphate pathway (PPP) were analyzed in AsPC-1 and PC-3 cells by targeted LC-MS/MS. Metabolite levels were normalized using the BCA protein assay (n = 3). Glucose (-), glucose deprivation; graph bars represent the mean ± sd. *, *p* < 0.05; **, *p* < 0.01; ***, *p* < 0.001; n.s., not significant.

**Figure 4 ijms-24-11969-f004:**
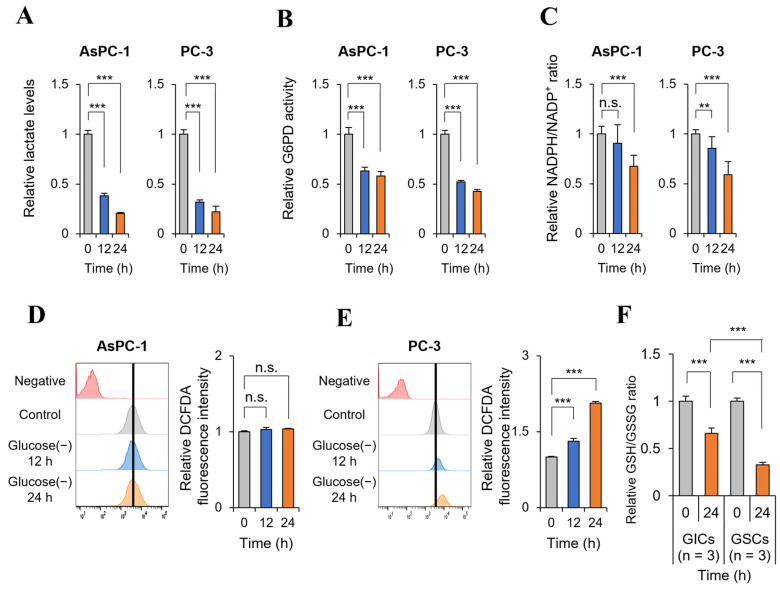
Glucose deprivation induces ROS production along with reductions of glycolysis, NADPH/NADP, and GSH/GSSG in cancer cells. (**A**) To test the effect of glycolysis by glucose deprivation, lactate levels were measured using a lactate assay kit in cells cultured under glucose deprivation conditions for 0, 12, and 24 h (n = 3). (**B**) To test the effect of glycolysis by glucose deprivation, glucose-6-phosphate dehydrogenase (G6PD) activity was measured using a G6PD assay kit in cells cultured under glucose deprivation conditions for 0, 12, and 24 h (n = 3). (**C**) To test the effect of NADPH production by glucose deprivation, the NADPH/NADP ratio was measured using a NADPH assay kit in cells cultured under glucose deprivation conditions for 0, 12, and 24 h (n = 3). (**D**,**E**) ROS levels in cells under glucose deprivation were analyzed by staining with 2,7-dichlorofluoroscin diacetate (DCFDA) and determined by flow cytometry (n = 3). Fluorescence intensity was quantified by the Geometric Mean in FlowJo software (v10.8.1). (**F**) The GSH/GSSG ratio was measured using a glutathione assay kit in cells cultured under glucose deprivation conditions for 0, 12, and 24 h (n = 3). Glucose (-), glucose deprivation; graph bars represent the mean ± sd. **, *p* < 0.01; ***, *p* < 0.001; n.s., not significant.

**Figure 5 ijms-24-11969-f005:**
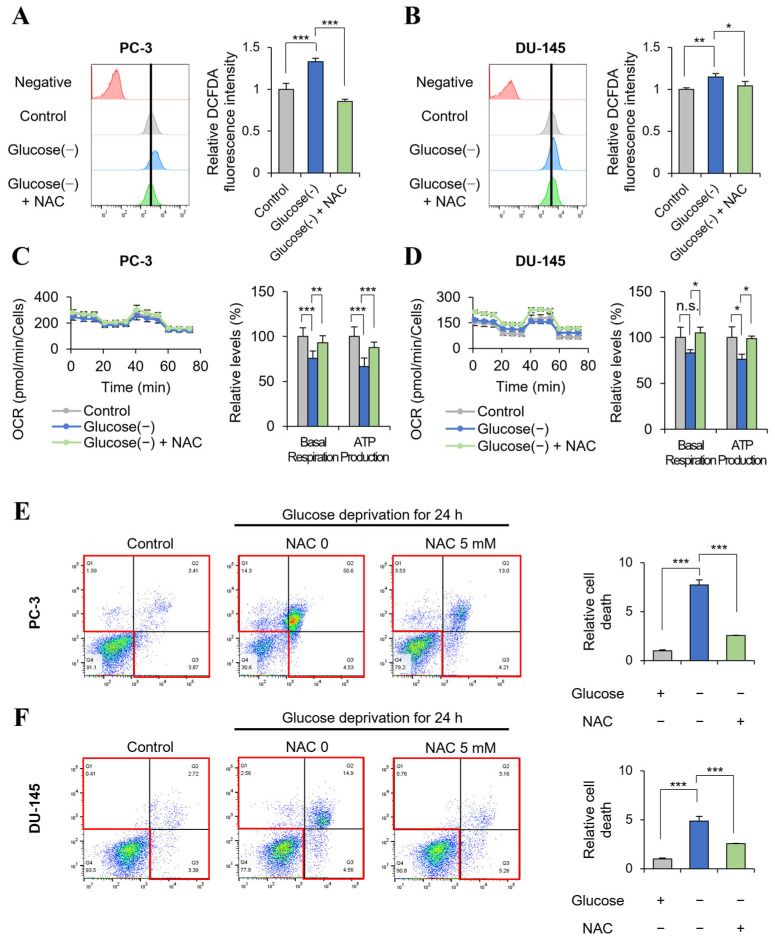
NAC treatment rescued ATP production by scavenging glucose deprivation-induced ROS in GSC. PC-3 and DU-145 (GSC) cells were pretreated with or without 5 mM NAC for 1 h prior to glucose starvation and co-treated with NAC for 12 h in glucose-free medium. (**A**,**B**) ROS levels were analyzed by staining with DCFDA and determined by flow cytometry (n = 3). (**C**,**D**) Basal respiration and ATP production were measured by XFe96 extracellular flux analysis in PC-3 cells cultured under glucose deprivation with or without NAC for 0 and 12 h (n = 6). (**E**,**F**) Cell death (red box)was measured by annexin V-FITC and PI staining in prostate cancer cells (GSCs) cultured under glucose deprivation conditions for 24 h and treated with or without NAC. The intensity was quantified by FlowJo software (v10.8.1) (n = 3). Glucose (-), glucose deprivation; graph bars represent the mean ± sd. *, *p* < 0.05; **, *p* < 0.01; ***, *p* < 0.001; n.s., not significant.

**Figure 6 ijms-24-11969-f006:**
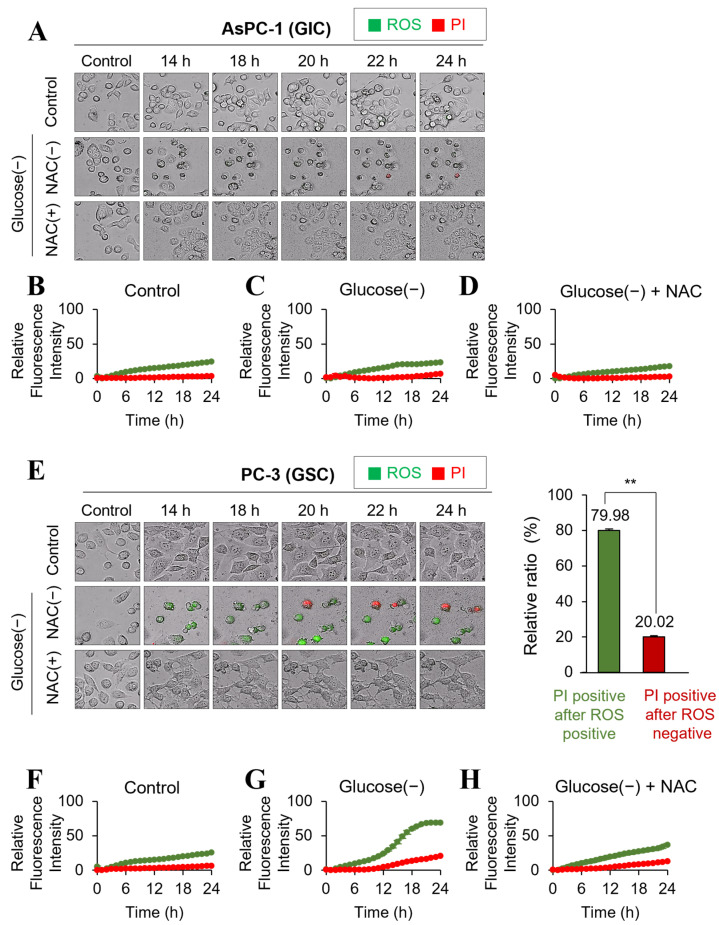
Cancer cell death was induced following an increase in ROS under glucose deprivation. Time-lapse fluorescence images of ROS (green) and PI (red) staining in AsPC-1 and PC-3 cells treated with or without NAC. (**A**) Real-time changes in ROS and PI levels in AsPC-1 cells were observed by live cell imaging using Operetta CLS. Images were acquired every hour for 24 h. The cells were stained with PI (1 µg/mL) and CellROX Green Reagent (5 µM) to detect cell death and oxidative stress. For NAC treatment evaluation, cells were pretreated with 5 mM NAC for 1 h prior to glucose starvation. (**B**–**D**) The relative fluorescence intensity of ROS and PI were quantified in AsPC-1 cells grown under normal glucose conditions (**B**), under glucose deprivation (**C**), and in cells pretreated with NAC before glucose deprivation (**D**) using Harmony 4.5 software. (**E**) Real-time changes in ROS and PI levels in PC-3 cells were observed as in (**A**). Cells were classified and quantified as described in the Materials and Methods section into the ‘PI positive after ROS positive’ group and the ‘PI positive after ROS negative’ group. The ratio of each group, represented as a percentage of the total PI-positive cells over the entire time period, was presented as the mean ± standard deviation (n = 3). ** *p* < 0.01. (**F**–**H**) The relative fluorescence intensity levels of ROS and PI were quantified in PC-3 cells grown under normal glucose conditions (**F**), under glucose deprivation (**G**), and in cells pretreated with NAC before glucose deprivation (**H**) using Harmony 4.5 software. Glucose (-), glucose deprivation. The picture was captured 20× and zoomed in 16× (**A**,**E**).

**Figure 7 ijms-24-11969-f007:**
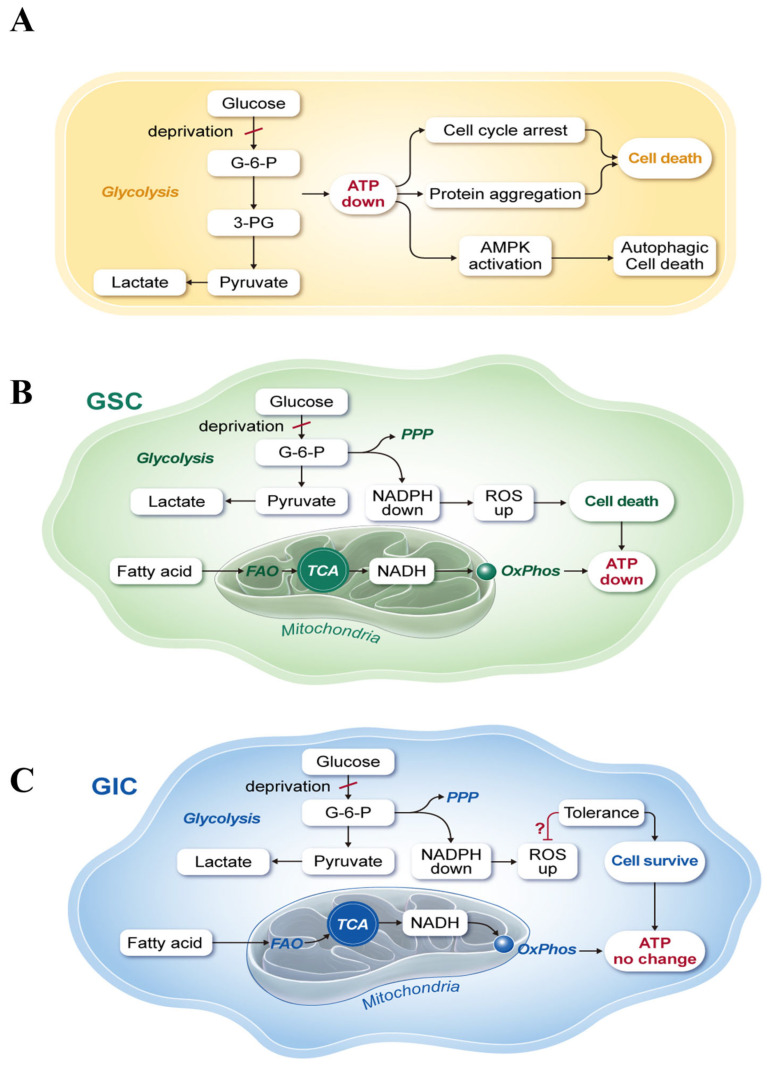
Schematic representation of the traditional and new glucose metabolism. (**A**) Traditional concept of glucose-dependent ATP production introduced by Warburg. The Warburg effect explains the role of glucose deprivation in cancer cell death as ATP depletion-mediated breakdown of cell homeostasis resulting in cell cycle arrest, protein aggregation, and autophagy activation. Recently, we proposed that the major metabolic pathway of ATP production in cancer cells is fatty acid oxidation (FAO) using fatty acids systemically supplied from the blood (Kim effect). (**B**) GSC are sensitive to glucose deprivation, which induces cell death by increasing ROS production through the downregulation of NADPH, resulting in decreased ATP production. (**C**) GIC did not show a decrease of ATP levels after glucose deprivation for 24 h. Although NADPH production was reduced by glucose deprivation through the depletion of PPP intermediates in both GIC and GSC, GIC did not show an increase in ROS levels mediated by a ROS tolerance mechanism such as the GSHR system. However, the exact mechanism of ROS tolerance in GIC remains to be elucidated (?).

## Data Availability

Not applicable.

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
