# Peer review of "Glucose Deprivation Induces Cancer Cell Death through Failure of ROS Regulation"

_ijms, 2023, doi:10.3390/ijms241511969_

Round 1
Reviewer 1 Report
In this paper “Glucose Deprivation Induces Cancer Cell Death through Failure of ROS Regulation”, the authors investigated whether the decrease in ATP production upon glucose deprivation is a cause or a result of cell death in GSCs. The paper presents sufficient background investigation and data, the results are meaningful and in line with the readers’ interest of IJMS. However, there are still some shortcomings that need to be further improved or explained.
Comments:
Q1. In abstract, there is a lack of description of the main research content of this paper.
Q2. In Figure 2, the resolution of the flow scatter plots is too low, please improve. The four quadrants are not properly divided of Fig. 2A. Besides, please explain the Please explain the presence of cells in the upper left quadrant of Fig. 2C. Annexin V should be changed to Annexin V-FITC. Which quadrant does Relative cell death represent the proportion of cells in?
Q3. The resolution of the flow scatter plots in other figures is also too low, please improve.
Q4. In Figure 4, There's a problem with the presentation of the relative (DCFDA) Fluorescence intensity. Flow cytometry is supposed to calculate the proportions of stained cells, how can you calculate the fluorescence intensity?
Q5. I think the ideas of the article are very novel, but some concepts are vague and lacking. Electron transport across the mitochondrial membrane is the process that generates ATP continuously, which involves NAD - and NADPH-mediated energy production. How can the authors suggest that the decrease in NADPH is causing an increase in ROS rather than a direct decrease in ATP synthesis.
Q6. Please provide diagrams of the mitochondrial or intracellular ATP and ROS production processes, and if necessary, the results of immunoblot hybridization experiments for key enzymes should be supplemented.
Author Response
Reviewer 1
In this paper “Glucose Deprivation Induces Cancer Cell Death through Failure of ROS Regulation”, the authors investigated whether the decrease in ATP production upon glucose deprivation is a cause or a result of cell death in GSCs. The paper presents sufficient background investigation and data, the results are meaningful and in line with the readers’ interest of IJMS. However, there are still some shortcomings that need to be further improved or explained.
Comments:
Q1. In abstract, there is a lack of description of the main research content of this paper.
A1. In abstract, we have added sentences for the object of study and conclusion.
“In previous work, we showed that cancer cell does not depend on glycolysis for ATP production, but it does on fatty acid oxidation. However, we found some cancer cells induced cell death after glucose deprivation along with decrease of ATP production. We investigated the different response of glucose deprivation with two types of cancer cells including glucose insensitive cancer cells (GICs) which does not change ATP levels, and glucose sensitive cancer cells (GSCs) which decreases ATP production in 24 h. ~~ Conclusion is clear that glucose deprivation-induced cell death is independent from ATP depletion-induced cell death.”
Q2. In Figure 2, the resolution of the flow scatter plots is too low, please improve. The four quadrants are not properly divided of Fig. 2A. Besides, please explain the Please explain the presence of cells in the upper left quadrant of Fig. 2C. Annexin V should be changed to Annexin V-FITC. Which quadrant does Relative cell death represent the proportion of cells in?
Q3. The resolution of the flow scatter plots in other figures is also too low, please improve.
A2&3. We deeply apologize for the poor resolution of the FACS data. We improved the resolution of flow scatter plots FACS data in Figure 2 with re-experimentation and quadrant adjustment to complement the data. The distribution of cells in the red box indicates the degree of cell death. We changed the legend.
Q4. In Figure 4, There's a problem with the presentation of the relative (DCFDA) Fluorescence intensity. Flow cytometry is supposed to calculate the proportions of stained cells, how can you calculate the fluorescence intensity?
A4. The fluorescence intensity of DCFDA positive cells was determined by the Geometric Mean in the FlowJo histogram based on gating the region forming the normal population while excluding the non-specific population with a negative control. We added this in the methods section.
Q5. I think the ideas of the article are very novel, but some concepts are vague and lacking. Electron transport across the mitochondrial membrane is the process that generates ATP continuously, which involves NAD - and NADPH-mediated energy production. How can the authors suggest that the decrease in NADPH is causing an increase in ROS rather than a direct decrease in ATP synthesis.
A5. Reviewer may be confused NADPH with NADH. NADH donates electron for ATP production. NADPH provides the reducing equivalents, usually hydrogen atoms, for biosynthetic reactions and the oxidation-reduction involved in protecting against the toxicity of reactive oxygen species (ROS), allowing the regeneration of glutathione (GSH)(Toxicology and Applied Pharmacology. 78 (3): 473–83). I think recent our review article may be helpful to understand current concepts of cancer metabolism (Seminars in Cancer Biology 2022 Nov;86(Pt 2):347-357).
Q6. Please provide diagrams of the mitochondrial or intracellular ATP and ROS production processes, and if necessary, the results of immunoblot hybridization experiments for key enzymes should be supplemented.
A6. ROS regulation is very sophisticated mechanism that cannot be explained by the expression level of enzymes. The reason is that the most enzymes contribute to the reversible reaction. That is why we showed the change of ROS level in Figures 4, 5, 6 instead of showing immuno blotting of key enzymes.
Reviewer suggests diagrams of the mitochondrial or intracellular ATP and ROS production processes, we disagree to integrate those diagrams in this paper. Because we have limited experimental data connecting dots between ROS and ATP production processes although each processes have many dots in multiple pathways. We discussed this issue in the end of discussion section as
“In this study, the reducing agent NAC rescued GSC from glucose deprivation-induced cell death (Figure 5). This implies that decreased ATP production is not the cause of cell death, but a result of cell death caused by failure of ROS regulation. ROS have dual and opposite roles in cancer, as they can induce cell proliferation to promote cancer growth, but can also have a tumor suppression effect by activating RCD programs such as apoptosis, necroptosis, and ferroptosis [22]. ROS levels are higher in cancer than in normal cells because cancer cells have a higher content of antioxidants [23]. The higher levels of ROS in cancer cells play critical roles in the activation of cell signaling and epithelial-mesenchymal transition [24], the activation of transcription factors such as Nrf2 [25], and the induction of lipid peroxidation products such as 4-hydroxy-2-nonenals [26], which can produce NADH for ATP production by aldehyde dehydrogenase [27]. However, increased ROS level also induces cancer cell death [22]. Cancer cell death is caused by glucose depletion [9] or glycolytic enzyme knockdown, as well as by chemotherapy or radiation therapy which is closely associated with induced ROS. Therefore, a role of ROS in anticancer therapies was proposed [28]. In response to stress related to ROS induction, antioxidants such as NAC promote cancer survival and progression, which is consistent with the data presented in Figure 5E,F [28].”

Reviewer 2 Report
In the research manuscript entitled “Glucose Deprivation Induces Cancer Cell Death through Failure of ROS Regulation” by Kang et al. the authors investigate glucose-deprivation induced cancer cell death. The major claim they make is that cancer cell death in glucose sensitive cells is regulated by ROS and not by ATP depletion. The topic is generally interesting, but not necessarily new. Other publications have investigated similar things in other cell types e.g. 10.1016/j.redox.2021.102065 , 10.3390/biomedicines9091101 or 10.3390/cells9071598 , just to name a few. The data presented within the manuscript looks robust, but is overinterpreted in their general meaning. The authors should emphasize on critical discussion of their data with other published work. In principle the data could be of interest to the readership of Molecular Sciences and it could be transformed into a valuable contribution, if a major revision was successful in addressing all the raised points as follows.
The following points should be addressed by the authors.
Major points:
The introduction from line 53-73 is a copy of the previous part and needs to be removed. The introduction is too short and needs a more thorough presentation of the previous work on this topic. Seven citations is definitely not all that one would expect to introduce this kind of research.
In Fig. 2A the FACS-gating seems inappropriate, as cell populations are not well separated by the acquired gates - especially for AsPC-1 cells. The samples need to be re-gated. Which fraction(s) were quantified in B and D. Is it the sum of AnnexinV-positive, PI-positive and PI/AnnexinV double positive? FloJo software is mentioned in the figure legend but this is not further explained in the methods section.
In Fig. 3 the authors should add bars to the asterisks indicating significant difference. That will help to make the comparisons clearer.
In Fig. 4 the authors claim: “Glucose deprivation induces ROS production through lactate, G6PD activity, NADPH reduction, and GSH/GSSG reduction in cancer cells.” I do not see any evidence underpinning this causal link. To make a causal conclusion the authors would have to reestablish normal lactate/ G6PD, etc. levels and show that this normalizes the ROS levels. Furthermore to this, lactate and G6PD activity is reduced upon glucose deprivation, which makes this phrase incorrect. I suggest rewriting this as a line of events/phenotypes without judgement about the dependencies.
In the discussion a more critical view should be shed on the author´s data by comparison to earlier studies. Also comparisons to other cell types should be made and discussed. Compare to metabolic analyses of cancer studies published in high impact journals.
In the statistics section the authors state that they used student´s t-test or Mann-Whitney-U test to test for statistical significance. Both tests are only valid for the comparison of two conditions. As in all experiments three conditions are present an ANOVA analysis - or non-parametric equivalent - is required as appropriate statistical tool. The whole statistics need to be recalculated accordingly.
Minor points:
Abstract, line 19: “… to compare to 0 time.” please rephrase to sound correctly.
Line 23/24: “The 2-fold higher the reduced/oxidized glutathione (GSH/GSSG) ratio in GICs than in GSCs contributed to the low level of ROS under glucose deprivation conditions.” The sentence is not correct. Please rephrase.
The FACS plots in Figure 2 A and C are poor in quality.
In Figure 4 panels D and E/F need to be swapped in the legend to fit the figure.
Line 168/169: ROS regulation is a complex mechanism because ROS producers and scavengers need to be balanced to achieve tolerance. A citation is lacking here. The abbreviation NAC needs to be introduced in the text when used for the first time.
Figure 5: What does “GF” stand for? Figure 6E: The authors should not use the expression “ROS-induced cell death” in the figure legend as it implicates a causal link, but at that stage the analysis can only visualize the marker. Please use judgement free expression such as “ROS/PI-double positive” What timepoint is shown in the bar diagram? What does the comparison of “ROS-induced cell death” with “direct cell death” mean? Why would you compare that? Replace “(e)” with capital “(E)”.
Discussion: “We found that the major source of ATP production in cancer is FAO in a process named the ‘Kim effect’ to avoid confusion with metabolic reprogramming [1].“ I don´t think that this kind of “self-glorification” is adequate, especially since this is based on a self-citation. In section “Statistics” change text to “All experiments were repeated at least three times” as in some cases (e.g. Fig. 1) n=5 is indicated.
Line 274: “mediated” is the wrong word. Apoptosis, necroptosis, and ferroptosis are modes of regulated cell death.
Line 343/344: “In summary, we conclude that ATP depletion by glucose deprivation is the result of cancer cell death caused by failure of ROS regulation.“ Weaken the claim by wording more cautiously. E.g we suggest…
The language in the manuscript is okay, but the style is not very appealing.
Author Response
Reviewer 2
In the research manuscript entitled “Glucose Deprivation Induces Cancer Cell Death through Failure of ROS Regulation” by Kang et al. the authors investigate glucose-deprivation induced cancer cell death. The major claim they make is that cancer cell death in glucose sensitive cells is regulated by ROS and not by ATP depletion. The topic is generally interesting, but not necessarily new. Other publications have investigated similar things in other cell types e.g. 10.1016/j.redox.2021.102065 , 10.3390/biomedicines9091101 or 10.3390/cells9071598 , just to name a few. The data presented within the manuscript looks robust, but is overinterpreted in their general meaning. The authors should emphasize on critical discussion of their data with other published work. In principle the data could be of interest to the readership of Molecular Sciences and it could be transformed into a valuable contribution, if a major revision was successful in addressing all the raised points as follows.
The following points should be addressed by the authors.
Major points:
Q1. The introduction from line 53-73 is a copy of the previous part and needs to be removed. The introduction is too short and needs a more thorough presentation of the previous work on this topic. Seven citations is definitely not all that one would expect to introduce this kind of research.
A1. We are sorry that we could not pay attention for our mistake of digital reformatting on submission. We have fixed it.
The short introduction reflected to the long discussion because there are few references about this subject. Even current top review article introduces that glucose is the major ATP source based on the Warburg effect. In discussion, we discussed the difference between ATP-depletion-induced cell death and glucose-depletion-induced cell death. A misunderstanding has arisen from confusing glucose-induced cell death with the same mechanism as cell death by ATP depletion based on the Warburg effect. In the discussion, we have discussed the difference.
“ATP depletion also induces apoptosis. In studies of energy metabolism, oligomycin treatment targeting OxPhos by inhibiting ATP synthase decreases ATP production and induces cell death [31]. Blocking OxPhos results in cell death after AMPK activation, ROS generation, and caspase activation despite normal glycolytic activity [31]. Oligomycin treatment under glucose deprivation conditions reduces ATP levels by >50% within 10 min [32]. However, cell death is not an immediate response to ATP depletion and is induced after 48 h [31]. ATP depletion may lead to multiple metabolic alterations of cell homeostasis before cell death occurs. Indeed, glucose deprivation-induced cell death is independent from ATP depletion-induced cell death.”
Q2. In Fig. 2A the FACS-gating seems inappropriate, as cell populations are not well separated by the acquired gates - especially for AsPC-1 cells. The samples need to be re-gated. Which fraction(s) were quantified in B and D. Is it the sum of AnnexinV-positive, PI-positive and PI/AnnexinV double positive? FloJo software is mentioned in the figure legend but this is not further explained in the methods section.
A2. We deeply apologize for the poor resolution of the FACS data. We improved the resolution of flow scatter plots FACS data in Figure 2 with re-experimentation and quadrant adjustment to complement the data. The distribution of cells in the red box indicates the degree of cell death.
Quantitative data in B and D are calculated as the sum of the Annexin V-positive, PI-positive, and Double-positive populations in the red boxes. This is described in 4.3. of the Methods section.
Q3. In Fig. 3 the authors should add bars to the asterisks indicating significant difference. That will help to make the comparisons clearer.
A3. Thank you for your suggestion. We have added bars to the asterisks.
Q4. In Fig. 4 the authors claim: “Glucose deprivation induces ROS production through lactate, G6PD activity, NADPH reduction, and GSH/GSSG reduction in cancer cells.” I do not see any evidence underpinning this causal link. To make a causal conclusion the authors would have to reestablish normal lactate/ G6PD, etc. levels and show that this normalizes the ROS levels. Furthermore to this, lactate and G6PD activity is reduced upon glucose deprivation, which makes this phrase incorrect. I suggest rewriting this as a line of events/phenotypes without judgement about the dependencies.
A4. First of all, glucose deprivation naturally resulted in the lactate depletion because cancer cell produces lactate only from glucose, called Warburg effect (Figure 4). We also have shown several times (Cancers 2020, 12(9), 2477; Semin Cancer Biol 2022 Nov;86(Pt 2):347-357). Glucose deprivation also naturally resulted in the reduced pentose phosphate pathway (PPP) because it is branched out from glycolysis. Reduction of glycolysis and PPP metabolism consequence to the reduction of NADPH production, which resulted in the increase of ROS (Nature Metabolism 1, 404–415, 2019). The change of metabolites in glycolysis and PPP are shown in the Figure 3. We have changed the legend following to the reviewer’s concerns as below.
Figure 4. Glucose deprivation induces ROS production along with reductions of glycolysis, NADPH/NADP and GSH/GSSG in cancer cells. (A) To test effect of glycolysis by glucose deprivation, lactate levels were measured using a lactate assay kit in cells cultured under glucose deprivation conditions for 0, 12, and 24 h (n = 3). (B) To test effect of glycolysis by glucose deprivation, glucose-6-phosphate dehydrogenase (G6PD) activity was measured using a G6PD assay kit in cells cultured under glucose deprivation conditions for 0, 12, and 24 h (n = 3). (C) To test effect of NADPH production by glucose deprivation, the NADPH/NADP ratio was measured using a NADPH assay kit in cells cultured under glucose deprivation conditions for 0, 12, and 24 h (n = 3).
Q5. In the discussion a more critical view should be shed on the author´s data by comparison to earlier studies. Also comparisons to other cell types should be made and discussed. Compare to metabolic analyses of cancer studies published in high impact journals.
A5. To address critical view on our data, at least experimental data sets must be available to compare to ours. However, we do not know exact reason why, but no data set is available for this. Even in the top journal research article did not show experimental data of ATP analysis after glucose deprivation, or lactate treatment as a fuel for cancer cell. That is why we mentioned this issue at the discussion section as,
“However, there are no experimental data demonstrating that glucose deprivation induces cell death by decreasing ATP production.”
There are reports that ATP depletion induced cancer cell death. They suggested several mechanisms of cell death, which we mentioned in the discussion section as,
“Several mechanisms of ATP deprivation-induced cell senescence or cell death have been proposed, including cell cycle arrest [10], increased ER stress [5], caspase 8 activation [6], increased protein aggregation [29], and AMPK-induced autophagic cell death [7, 30] among others.”
In other cancer cells, GIC cancer cells has been published twice (Cancers 12, 9 (2020):2477; Seminars in Cancer Biology 2022 Nov;86(Pt 2):347-357), and GSC cancer cells report in this paper Figure S1. The result was in 2.1 section.
“However, in some cancer cell lines, including glioma cells (U87 and T98G), breast cancer cells (MDA-MB-231 and MCF-7), and colon cancer cells (KM12 and HT-29), glucose dep-rivation for 12 h decreased ATP production by 20–80% compared with the control cells cultured with high glucose (Figure 1C,D, & Figure S1). These cells were grouped as GSC.”
Q6. In the statistics section the authors state that they used student´s t-test or Mann-Whitney-U test to test for statistical significance. Both tests are only valid for the comparison of two conditions. As in all experiments three conditions are present an ANOVA analysis - or non-parametric equivalent - is required as appropriate statistical tool. The whole statistics need to be recalculated accordingly.
A6. Thank you for your comments. We did not perform statistics carefully. By following your suggestion, ANOVA test updated our results in Figures 1~6, and changed the Figures.
We revised 4.11 statistical analysis in the method section.
“All statistical analyses were performed using the GraphPad Prism 9 software (GraphPad Software Inc., San Diego, CA, USA). Statistical significance was analyzed with a one-way analysis of variance (ANOVA). P-values are denoted as follows: *P < 0.05, **P < 0.01, ***P < 0.001, ****P < 0.0001.”
Minor points:
Q7. Abstract, line 19: “… to compare to 0 time.” please rephrase to sound correctly.
A7. We changed to “Glucose deprivation-induced cell death in GSCs by more than 2-fold after 12 h and by up to 10-fold after 24 h accompanied by decreased ATP production to compare to the control (cultured in glucose).”
Q8. Line 23/24: “The 2-fold higher the reduced/oxidized glutathione (GSH/GSSG) ratio in GICs than in GSCs contributed to the low level of ROS under glucose deprivation conditions.” The sentence is not correct. Please rephrase.
A8. We have changed to “The 2-fold higher ratio of reduced/oxidized glutathione (GSH/GSSG) in GIS than in GSC correlates closely with 2-fold lower ROS levels under glucose starvation conditions.”
Q9. The FACS plots in Figure 2 A and C are poor in quality.
A9. Figure 2 is replaced with a new Data supplemented by re-experimentation, with red boxes added to show quantified dead cell populations.
Q10. In Figure 4 panels D and E/F need to be swapped in the legend to fit the figure.
A10. We are sorry for the mistake. We fixed the mistakes.
Q11. Line 168/169: ROS regulation is a complex mechanism because ROS producers and scavengers need to be balanced to achieve tolerance. A citation is lacking here. The abbreviation NAC needs to be introduced in the text when used for the first time.
A11. NAC was used first time in the abstract which missed the “(NAC)”. We introduced one more time at p7. We have a reference for this [12].
Q12. Figure 5: What does “GF” stand for? Figure 6E: The authors should not use the expression “ROS-induced cell death” in the figure legend as it implicates a causal link, but at that stage the analysis can only visualize the marker. Please use judgement free expression such as “ROS/PI-double positive” What timepoint is shown in the bar diagram? What does the comparison of “ROS-induced cell death” with “direct cell death” mean? Why would you compare that? Replace “(e)” with capital “(E)”.
A12. We appreciate your detailed review and sincerely apologize for any confusion caused by our previous expressions. We added the meaning of GF in Figure 5; GF stands for glucose free.
In Figure 6E, we replaced the term 'ROS-induced cell death' with ' PI positive after ROS positive,' and 'direct cell death' with 'PI positive after ROS negative' to accurately reflect our findings. Furthermore, we have included detailed analysis methods in the Materials and method section.
" Real-time live cell images were acquired using the Operetta CLS (Perkin Elmer), capturing bright-field and fluorescent images every hour from 0 to 24 h, as previously described [36]. For each time point, nine field images were taken per well, and a minimum of 1 × 104 cells were analyzed per well. High content analysis was performed using Harmony 4.5 software (Perkin Elmer). The cell count and fluorescent intensity for each cell were quantified in the images at each time point. Intensity values higher than those of the control cells were considered as positive signals. Cells that exhibited ROS-postive signals prior to the appearance of PI-positive signals were classified into the ‘PI positive after ROS positive’ group. Cells that exhibited only PI-positive signals without showing preceding ROS-positive signals were classified into the 'PI positive after ROS negative' group.”
In Figure 6, legend is changed to:
“Cells were classified and quantified as derived in Matierials and Methods section, into the ‘PI positive after ROS positive’ group and the 'PI positive after ROS negative' group. The ratio of each group, represented as a percentage of the total PI-positive cells over the entire time point, was presented as the mean ± standard deviation (n=3).”
Q13. Discussion: “We found that the major source of ATP production in cancer is FAO in a process named the ‘Kim effect’ to avoid confusion with metabolic reprogramming [1].“ I don´t think that this kind of “self-glorification” is adequate, especially since this is based on a self-citation. In section “Statistics” change text to “All experiments were repeated at least three times” as in some cases (e.g. Fig. 1) n=5 is indicated.
A13. “self-glorification” criticism is misunderstanding of intention of using that terminology in cancer energy metabolism. We have looked proper simple abbreviation to explain a new paradigm of cancer energy metabolism which is composed of our 3 major new findings. First, cancer cell does not depend on glycolysis for ATP production (below list of ref. No:3,5,8,14,15,18,20,23,24). Second, cancer mitochondria is not damaged but functional and active. Oxidative phosphorylation is active in cancer cell, which plays a key role of ATP production in mitochondria (below list of ref. No:13,19,20,23,24). Third, cancer cell depends on fatty acid oxidation for ATP production using fatty acids supplied not from microenvironment but from blood (below list of ref. No: 15,21,22,24). All three new findings are totally contradicted to the explanation of energy metabolism using Warburg effect. Especially the third discovery is very important but there is a clear difference between metabolic reprogramming theory and ours. The metabolic reprogramming theory also suggests that cancer cell can use fatty acid oxidation for ATP generation under the limited condition such as microenvironment. However, we found that the fatty acids for fatty acid oxidation in cancer cell are systemically supplied from blood vessel by dietary food. This big difference has to be named concisely. Please take a look carefully our recent review article (24. Seminars in Cancer Biology 2022 Nov;86(Pt 2):347-357). Then you will see how important it is because recent top review articles (Trends in Cell Biology, 2023 Apr 26;S0962-8924(23)00070-3) still make a note that glycolysis is the main source for ATP production in cancer cell. Therefore, we proposed “Kim effect” followed by previous style of naming “Warburg effect” to send a message that paradigm of cancer energy metabolism must be changed. Glucose deprivation-induced cell death is traditionally explained as the result of decreased ATP production based on the “Warburg effect”, which proposes that glycolysis is the major metabolic pathway for ATP production (Figure 7A). However, our new finding explains glucose deprivation caused cell death by failure of ROS regulation, based on “Kim effect”. Therefore, we believe that we are deserved to have a right to name it as “Kim effect”. To come up with this theory, we have investigated cancer energy metabolism over 10 years and published over 20 papers as a main investigator. The list of our publications is followed.
<The list of our publications related with cancer energy metabolism>
- Cancer metabolism: targeting cancer universality. Arch. Pharm. Res. 2015 38(3):299-301
- Cancer Metabolism: Strategic Diversion from Targeting Cancer Drivers to Targeting Cancer Suppliers. Biomol Ther 2015 23(2):99-109
- Aldehyde dehydrogenase inhibition combined with phenformin treatment reversed NSCLC through ATP depletion Oncotarget, 2016 2;7(31):49397-49410
- Glutaminase 1 inhibition reduces thymidine synthesis in NSCLC. Biochem Biophys Res Commun (2016) 477(3):374-82.
- Aldehyde dehydrogenase is served for cancer energy metabolism. Exp Mol Med 2016 48, e272
- Migration and invasion of drug-resistant lung adenocarcinoma cells are dependent on mitochondrial activity. Exp Mol Med. (2016) 48(12):e277.
- Dual targeting of glutaminase 1 and thymidylate synthase elicits death synergistically in NSCLC. Cell Death and Disease, 2016 Dec:7(12):e2511
- Snail reprograms glucose metabolism by repressing PFKP allowing cancer 2 cell survival under metabolic stress. Nature Communication , 2017 8:17374 IF12.124
- A clinical drug library screen identifies clobetasol propionate as an NRF2 inhibitor with potential therapeutic efficacy in KEAP1 mutant lung cancer. Oncogene. 36(37) 2017 5285–5295
- Targeting Mitochondrial Oxidative Phosphorylation Abrogated Irinotecan Resistance in NSCLC
Scientific Reports 15707 (2018)
- Regulation of bioenergetics through dual inhibition of aldehyde dehydrogenase and mitochondrial complex I suppresses glioblastoma tumorspheres. Neuro Oncol. 2018 Jun 18;20(7):954-965.
- Targeting cancer energy metabolism: a potential systemic cure for cancer. Arch.Pharm.Res 42.2 (2019): 140-149.
- Loss of SLC25A11 causes suppression of NSCLC and melanoma tumor formation. Ebiomedicine 40 (2019): 184-197.
- Gossypol Suppresses Growth of Temozolomide-Resistant Glioblastoma Tumor Spheres. Biomolecules 9.10 (2019): 595.
- Gastric cancer depends on aldehyde dehydrogenase 3A1 for fatty acid oxidation. Scientific Reports 9.1 (2019)
- Glutathione peroxidase-1 regulates adhesion and metastasis of triple negative breast cancer cells via FAK signaling. Redox biology 29 (2020): 101391.
- Snail augments fatty acid oxidation by suppression of mitochondrial ACC2 during cancer progression. Life Science Alliance 3.7 (2020).
- The Combination of Loss of ALDH1L1 Function and Phenformin Treatment Decreases Tumor Growth in KRAS‐Driven Lung Cancer. CANCERS 12.6 (2020): 1382.
- Targeting Oxidative Phosphorylation Reverses Drug Resistance in Cancer Cells by Blocking Autophagy Recycling. Cells 9, 9 (2020):2013.
- Oxoglutarate Carrier Inhibition Reduced Melanoma Growth and Invasion by Reducing ATP Production. Pharmaceutics (2020)12,1128
- ATP Production Relies on Fatty Acid Oxidation Rather than Glycolysis in Pancreatic Ductal Adenocarcinoma. CANCERS 12, 9 (2020):2477.
- Overall survival of pancreatic ductal adenocarcinoma is doubled by Aldh7a1 deletion in the KPC mouse. Theranostics. 2021 Jan 19;11(7):3472-3488.
- Combinatorial Therapeutic Effect of Inhibitors of Aldehyde Dehydrogenase and Mitochondrial Complex I, and the Chemotherapeutic Drug, Temozolomide against Glioblastoma Tumorspheres. Molecules. 2021 Jan 8;26(2):282 (IF 4.411)
- Cancer depends on fatty acids for ATP production: a possible link between cancer and obesity. Semin Cancer Biol. 2022 Nov;86(Pt 2):347-357.
In section “Statistics”, we changed information adequately.
- Line 274: “mediated” is the wrong word. Apoptosis, necroptosis, and ferroptosis are modes of regulated cell death.
A14. We changed to, “ROS induce the regulated cell death (RCD) program such as apoptosis, necroptosis, and ferroptosis”
Q15. Line 343/344: “In summary, we conclude that ATP depletion by glucose deprivation is the result of cancer cell death caused by failure of ROS regulation.“ Weaken the claim by wording more cautiously. E.g we suggest…
A15. We changed the sentences. “These findings suggest that ATP depletion resulting from glucose deprivation is not the cause of cancer cell death, but the result of cancer cell death caused by failure of ROS regulation. Indeed, glucose deprivation-induced cell death is independent from ATP depletion-induced cell death.”

Reviewer 3 Report

Required improvement
Author Response
Reviewer 3
Major& Minor comments:
Q1. Abstract: Author needs to give background of work why this study is important and basic pathophysiology when addressing the work in abstract.
A1. Thank you for your comments. We have added the background of this study in the abstract.
“Abstract: In previous work, we showed that cancer cell does not depend on glycolysis for ATP production, but it does on fatty acid oxidation. However, we found some cancer cells induced cell death after glucose deprivation along with decrease of ATP production. We investigated the dif-ferent response of glucose deprivation with two types of cancer cells including glucose insensitive cancer cells (GIC) which does not change ATP levels, and glucose sensitive cancer cells (GSC) which decreases ATP production in 24 h. Glucose deprivation-induced cell death in GSC by more than 2-fold after 12 h and by up to 10-fold after 24 h accompanied by decreased ATP production to compare to the control (cultured in glucose). Glucose deprivation decreased the levels of metabolic intermediates of the pentose phosphate pathway (PPP) and the reduced form of nicotinamide adenine dinucleotide phosphate (NADPH) in both GSC and GIC. However, glucose deprivation increased reactive oxygen species (ROS) only in GSC, suggesting that GIC have a higher tolerance for decreased NADPH than GSC. The 2-fold higher ratio of reduced/oxidized glutathione (GSH/GSSG) in GIS than in GSC correlates closely with 2-fold lower ROS levels under glucose starvation conditions. Treatment with N-acetylcysteine (NAC) as a precursor to the biologic an-tioxidant glutathione restored ATP production by 70% and reversed cell death caused by glucose deprivation in GSC. The present findings suggest that glucose deprivation-induced cancer cell death is not caused by decreased ATP levels, but rather triggered by failure of ROS regulation. Conclusion is clear that glucose deprivation-induced cell death is independent from ATP deple-tion-induced cell death.”
Q2. Line 19; 0 time change to 0 h.
A2. We have changed to “to compare to the control (cultured in glucose).”
Q3. Line 29; Next to “ROS regulation” needs to add “by the antioxidant system”.
A3. We have added the “by the antioxidant system” in the abstract.
Q4. Introduction: Author needs to give background of work why this study is important and basic pathophysiology at the start of introduction. Also, needs to brief about the importance of study in cancer cells death. Also, Author needs to brief about the migration of cancer cells in glucose deprivation.
A4. Thank you for your comments. The short introduction reflected to the long discussion because there are few references about this subject. Even current top review article introduces that glucose is the major ATP source based on the Warburg effect. In discussion, we discussed the difference between ATP-depletion-induced cell death and glucose-depletion-induced cell death. A misunderstanding has arisen from confusing glucose-induced cell death with the same mechanism as cell death by ATP depletion based on the Warburg effect. In the discussion, we have discussed the difference.
“ATP depletion also induces apoptosis. In studies of energy metabolism, oligomycin treatment targeting OxPhos by inhibiting ATP synthase decreases ATP production and induces cell death [31]. Blocking OxPhos results in cell death after AMPK activation, ROS generation, and caspase activation despite normal glycolytic activity [31]. Oligomycin treatment under glucose deprivation conditions reduces ATP levels by >50% within 10 min [32]. However, cell death is not an immediate response to ATP depletion and is induced after 48 h [31]. ATP depletion may lead to multiple metabolic alterations of cell homeostasis before cell death occurs. Indeed, glucose deprivation-induced cell death is independent from ATP depletion-induced cell death.”
Q5. Line 53- 70: These lines are repeated and need to change also rewrite the introduction from the perspective of study.
A5. We are sorry about the mistake of formatting error. We have fixed the problem.
Q6. Figure 1: Figure 1 legend needs rewrite like figure 2 and to mention for what this figure done using which cell lines, chemicals time period of treatment and subsection (a,b,c,d) what it is representing for each images.
A6. Thank you for your comments. We have changed the legend
“(A,B) Basal respiration and ATP production were measured by XFe96 extracellular flux analysis in pancreatic cancer cells [glucose insensitive cells (GIC): AsPC-1 and Panc-1] after glucose deprivation for 0, 12, and 24 h (AsPC-1, n = 5; Panc-1, n = 4). (C,D) Basal respiration and ATP production were measured by XFe96 extracellular flux analysis in prostate cancer cells [glucose sensitive cells (GSC): PC-3 and DU-145] after glucose deprivation for 0, 12, and 24 h (PC-3, n = 4; DU-145, n = 4). Glucose (-), glucose deprivation; graph bars represent the mean ± sd. *, p < 0.05; **, p < 0.01; ***, p < 0.001; n.s., not significant.”
Q7. Line 85-105; Result section needs to explain how the results came and what the author's interpretation, how this result is correlated with the other experiments within this study and can explain about the pros and cons of results in relevance with the study objective. In the discussion section, the author’s results can correlate with other groups of researchers done in the same field and provide proof of this experiment interpretation.
A7. Thank you for your comments. We have discussed the reason why Warburg effect misunderstood the glycolytic effect in the discussion p215~234. It is because of limit of experimental condition.
Also, we have discussed the reason why contradictory conclusion has happened between Warburg and ours in the discussion (p247~316). It is because they confused glucose deprivation-induced cell death (p247~281) and ATP depletion-induced cell death (p289~316). We have discussed in the discussion section. Indeed, glucose deprivation-induced cell death is independent from ATP depletion-induced cell death.
Q8. Figure 2: Scale of bar graph 2b needs to modify for clear representation of ns.
A8. Thank you. We have changed the Figure 2.
Q9. Line 129-130: Lactate also decreased in 1hr, then what is interpretation for this condition needs to include in this sentence.
A9. This means that lactate drops rapidly in just one hour. Therefore, if cancer cells are relying on glycolysis to produce ATP, we should see a rapid decline in ATP production. However, ATP declines much later. There fore we mentioned it by following sentence,
“However, no changes in ATP levels were observed at 6 h in GSCs and GICs (Figure 3A,B & Figure S3C).”
Q10. Line 131: Author mentioned as “no change in ATP levels were observed in 6 h in GSC”, but Figure 3A&B shows no change at 3& 6 h compared to control.
A10. Thank you. We changed it, “at 3 and 6 hs”.
Q11.Line 174: Check whether it is GSH/GSSG or GSSHR? Authors need to mention all abbreviation at initial usage.
A11. We are sorry for the typos. We have fixed it.
Q12. Supplementary Results: Author needs to include the results of supplementary results in the result section and requires discussion in correlation with the main results.
A12. Thank you. We missed citation of supplementary figures 4 and 5. We have corrected them.
“In the presence of high glucose levels, the cellular GSH/GSSHR ratio did not differ between GIC (AsPC-1, SNB-75, and SK-OV-3) and GSC (KM12, PC-3, and MDA-MB-231), whereas under glucose deprivation conditions, the GSH/GSSG ratio was 2-fold higher in GIC than in GSC (Figure 4F & Figure S4). Therefore, a higher GSH/GSSG ratio in GIC than in GSC may contribute to ROS tolerance.”
“Glucose deprivation caused a 20–30% increase in ROS levels in PC-3 and DU-145 cells (GSC) (Figure 5A,B) and other GSC cells (U87, MDA-MB0231, KM12) (Figure S5A,C,E). Treatment with 5 mM NAC for 12 h decreased ROS production to control levels (Figure 5A,B). In addition, NAC restored ATP production to 70% of the control in PC-3 and DU-145 cells (GSC) (Figure 5C,D) and to 70–90% of the control in GSC (U87, MDA-MB0231, KM12) (Figure S5B,D,F).”
Q13. Line262: Abbreviation needs to elobrate on first usage. Also, in previously mentioned GSHR or GSSHR required abbreviation & Check for GSH/GSSG.
A13. Thank you. We have corrected it. “ratio of reduced/oxidized glutathione (GSH/GSSG)”.
Q14.Line 453: “ROS regulation” needs to add “by the antioxidant system”.
A14. We added the sentence.

Round 2
Reviewer 1 Report
The resolution of relevant figures in this article has not been improved, which means that the author has no original FCM data . The explanation of several points fell short of expectations.
Reviewer 2 Report
The authors have addressed all raised points. The introduction is still very short which leaves room for improvement.